# Regularized Diffusion Modeling for CAD Representation Generation

## Abstract

Computer-Aided Design (CAD) has significant practical value in various industrial applications. However, achieving high-quality and diverse shape generation, as well as flexible conditional control, remains a challenge in the field of CAD model generation. To address these issues, we propose CADiffusion, a diffusion-based generative model with a hierarchical latent representation tailored to the complexities of CAD design processes. To enhance the performance and reliability of the model in generating accurate CAD models, we have developed a specialized decoder with regularization strategies that navigate through the noise space of the diffusion model, smoothing the results. This approach not only improves the diversity and quality of the generated CAD models but also enhances their practical applicability, marking a significant advancement in the integration of generative models and automated CAD systems.

## 1 Introduction

In the field of 3D computer vision, exploring the generation of 3D shapes has emerged as a prominent issue in recent times. Various research studies have been conducted that encompass explicit representations such as point clouds Yang et al. (2019); Cai et al. (2020); Mo et al. (2019), polygon meshes Groueix et al. (2018); Wang et al. (2018); Nash et al. (2020), voxel grids Liao et al. (2018); Li et al. (2017), and implicit representations Park et al. (2019); Mescheder et al. (2019); Chen et al. (2020) such as neural radiance fields (NeRFs) and signed distance functions (SDFs). Despite the significant progress achieved by these methodologies, the effectiveness of generating 3D models for practical applications remains unsatisfactory, mainly due to constraints related to data availability and modeling capabilities of the models. In contrast to other forms of 3D data, Computer-Aided Design (CAD) data holds immense practical value and finds applications across various industrial domains, ranging from automotive and aerospace to manufacturing and architectural design. CAD software serves as the cornerstone for creating 3D shapes in these domains, facilitating intricate design processes, and streamlining manufacturing workflows. Therefore, exploring the generation of CAD models is highly significant as it has the potential to innovate many existing production processes. Our work primarily explores representations and the corresponding generative model structures that are more suitable for CAD generation.

Despite some existing research on the generation of CAD data, challenges persist to enhance the quality and diversity of the generated shapes. Current methodologies face challenges in generating highly diverse CAD shapes, primarily due to limitations in supporting the intricacies of CAD spatial modeling. DeepCAD Wu et al. (2021) generates CAD commands directly without modeling a space suitable for CAD generation. Moreover, the exploration of generative models remains limited to those such as GANs and autoregressive transformers Xu et al. (2022; 2023). Beyond the issues of quality and diversity, existing methods also struggle to ensure the consistency and realism of the generated results with the input conditions in conditional CAD generation.

To address these challenges, we introduce a hierarchical implicit space and, on top of it, we propose CADiffusion, a diffusion-based generative model with a latent representation structured in a tree logic. Most modern CAD design tools employ a "Sketch and Extrude" style workflow, where designers first draw loops of 2D curves as outer and inner boundaries to create 2D profiles, then extrude the 2D profiles to 3D shapes, and finally add or subtract 3D shapes to build complex CAD models. Therefore, a hierarchical representation perfectly aligns with the inherent logic of CAD itself. This

hierarchical structure also offers effective design control at different levels of the hierarchy. When modeling the latent space for CAD, we also need to consider the external, visible compositional logic of CAD. In our approach, the CAD data is divided into three distinct hierarchical levels, with each level employing a VAE to obtain latent representations. These representations are then organized into a tree logic latent structure.

To learn the probability distribution in the proposed latent space, we leverage diffusion models, which have recently achieved significant success in various 2D generation tasks. We find that diffusion modeling method also exhibits high-quality and highly diverse generation capabilities in CAD data. After using the diffusion model to fit the CAD data, to convert the CAD latent generated by the diffusion model into CAD models accurately and efficiently, we designed a corresponding decoder along with regularization terms. This involves navigating through the diffusion model's noise space and smoothing the outcomes of the sampled noise. This process ensures that the decoder can translate any sample from the Gaussian noise space into a reasonable CAD model. The regularization approach not only enhances the decoder's ability to handle variations but also contributes to the overall stability and reliability of CAD model generation in an automated setting. By integrating this regularized training methodology, we can bridge the gap between generating realistic data and the latent diffusion modeling, ensuring that the enhancements in generative model technology translate effectively into practical improvements in CAD systems. We evaluate the effectiveness of CADiffusion on benchmark datasets and show that it outperforms baseline approaches in a variety of metrics.

Therefore, our contributions can be summarized as follows: **1)** We are the first to explore the use of diffusion models for CAD generation, and have designed corresponding models and representations. **2)** We have introduced a new regularization strategy specifically for CAD latent diffusion models, enabling the decoder to produce more reasonable and higher quality results. **3)** Our CAD generation model achieves state-of-the-art performance, surpassing previous methods.

## 2 RELATED WORK

### 2.1 3D GENERATIVE MODELS

In recent years, significant attention has been directed towards the development of generative models for 3D shapes. Many existing approaches generate 3D shapes discretely, employing representations such as voxelized shapes Liao et al. (2018); Li et al. (2017), point clouds Yang et al. (2019); Cai et al. (2020); Mo et al. (2019), polygon meshes Groueix et al. (2018); Wang et al. (2018); Nash et al. (2020), and implicit signed distance fields Park et al. (2019); Mescheder et al. (2019); Chen et al. (2020). Despite their prevalence, these models often produce shapes with noise, limited geometric sharpness, and lack direct user editability. To address these limitations, recent research has focused on neural network architectures that generate 3D shapes through sequences of geometric operations. CSGNet Sharma et al. (2018), for instance, infers Constructive Solid Geometry (CSG) operations from voxelized shape inputs, while UCSG-Net Kania et al. (2020) enhances the inference process without relying on ground truth CSG trees. In addition, some approaches use domain-specific languages (DSLs) Mo et al. (2019); Jones et al. (2020) to synthesize 3D shapes. For instance, ShapeAssembly by Jones *et al.* Jones et al. (2020). introduces a DSL that constructs 3D shapes using hierarchical and symmetrical cuboid proxies, which can be generated through a variational autoencoder.

### 2.2 CAD GENERATION

Early research focused on direct CAD modeling without using any supervision from CAD modeling sequences. A common theme is to construct parametric curves Wang et al. (2020) and surfaces Sharma et al. (2020) with fixed Smirnov et al. (2021) or arbitrary topology for sketches Willis et al. (2021c) and solid models Wang et al. (2022); Guo et al. (2022); Jayaraman et al. (2022). In recent years, the availability of large-scale parametric CAD datasets has allowed learning-based methods to take advantage of data from CAD modeling sequences Willis et al. (2021b); Wu et al. (2021); Xu et al. (2022) and sketch constraints Seff et al. (2020) to generate engineering sketches and solid models. The resulting sequences can be processed using a solid modeling kernel to acquire editable parametric CAD files containing 2D engineering sketches Willis et al. (2021c); Para et al. (2021); Ganin et al. (2021); Seff et al. (2022) or 3D CAD shapes Wu et al. (2021); Xu et al. (2022). Furthermore, the

generation process may be influenced by the target B-rep Willis et al. (2021b); Xu et al. (2021), sketches Li et al. (2020); Seff et al. (2022), images Ganin et al. (2021), voxel grids Lambourne et al. (2022), or point clouds Uy et al. (2021), occasionally with sequence guidance Ren et al. (2022); Li et al. (2023).

More recently, there have been some advancements in the field of CAD model generation. Deep-CAD Wu et al. (2021) directly generates CAD commands without modeling the data representations first. CAD models are graphically and geometrically complex, and generating commands can lead to overly simplified results. Consequently, the outcomes from DeepCAD are generally quite simple. Subsequent efforts, such as those by SkexGen Xu et al. (2022) and HNC Xu et al. (2023), have employed autoregressive transformer models. Their use of discrete formats excessively compresses the representations, failing to effectively capture the intrinsic logic of CAD data. The representations utilized by these methods are either too redundant or semantically sparse, which impairs the generative model's performance in fitting them. Compared to other 3D data, CAD possesses parametric characteristics, and suitable representations and models for it are still under exploration.

### 2.3 Diffusion Models

Diffusion Probabilistic Models (DPMs) Sohl-Dickstein et al. (2015); Ho et al. (2020), commonly referred to as diffusion models, have emerged as a robust class of generative models. Unlike previous leading generative models such as the Generative Adversarial Network Goodfellow et al. (2020), Variational Autoencoder (VAE) Kingma & Welling (2014), and flow-based generative models Rezende & Mohamed (2015), diffusion models exhibit notable advantages in terms of training stability and generative diversity Croitoru et al. (2023). They have shown promising performance in image Ho et al. (2020); Dhariwal & Nichol (2021); Nichol et al. (2022); Rombach et al. (2022) and speech Chen et al. (2021); Kong et al. (2021) synthesis. In particular, diffusion model-based approaches have shown remarkable results in text-to-image synthesis Ramesh et al. (2022); Rombach et al. (2022); Saharia et al. (2022). In the realm of 3D computer vision, several studies have embraced diffusion models for generative 3D modeling Luo & Hu (2021); Zhou et al. (2021); Zeng et al. (2022). For example, PVD Zhou et al. (2021) used diffusion models to create 3D shapes using a point-voxel 3D representation. Luo *et al.* Luo & Hu (2021) considered points in point clouds as particles within a thermodynamic system with a heat bath. LION Zeng et al. (2022) introduced a VAE framework with hierarchical diffusion models in latent space. Similar attempts Chou et al. (2023); Cheng et al. (2023) have also applied diffusion models to the generation of SDFs. However, no exploration of diffusion models has been made on CAD data. We are the first to attempt using diffusion models to generate CAD data and have achieved very promising results.

## 3 Method

### 3.1 Preliminaries: Hierarchical CAD Representation

CAD (Computer-Aided Design) models are inherently hierarchical because of the nature of the objects they represent. This hierarchical structure is essential to accurately represent and manipulate engineering designs, mechanical components, and architectural plans. Thus, in our approach, we employ a hierarchical representation for CAD that builds on the foundations laid by SkexGen Xu et al. (2022) and HNC Xu et al. (2023), which themselves are extensions of the pioneering work of TurtleGen Willis et al. (2021a) and DeepCAD Wu et al. (2021).

**CAD Representation:** Similar to HNC Xu et al. (2023), we conceptualize a CAD model as a tree, where it is organized into three levels: Solid, Profile, and Loop. At the lowest level, a "loop" represents the basic connected curve in the model. It is composed of a set of lines, arcs, and circles. Each such primitive is defined by two, three, or four xy-coordinates, $L = \{(x_1, y_1), (x_2, y_2), \langle \text{SEP} \rangle, (x_3, y_3), \ldots\}$. Moving up the hierarchy, a "profile" defines a closed area on a sketch plane. It is constructed from a group of 2D bounding boxes. Each bounding box encompasses multiple loop elements that form part of the sketch, $P = \{(x_i, y_i, w_i, h_i)\}_{i=1}^{N_i^{\text{loop}}}$. $(x_i, y_i)$ is the bottom-left corner of the bounding box. $(w_i, h_i)$ is the width and height. Finally, at the top level, a "solid" represents a set of extruded profiles.

These extruded profiles are combined to form the entire 3D model, $S = \{(x_j, y_j, z_j, w_j, h_j, d_j)\}_{j=1}^{N_j^{\text{profile}}}$. The solid is described by a set of 3D bounding box parameters, providing a comprehensive representation of the volumetric aspects of the model. $j$ is the index of the $N_j^{\text{profile}}$ extruded profiles within a model. $(x_j, y_j, z_j)$ is the bottom-left corner of the bounding box and $(w_j, h_j, d_j)$ is its dimension.

**Hierarchical Latent Representation:** We use an adaptation of vector quantized VAE (VQ-VAE) van den Oord et al. (2017) consisting of a transformer encoder $E$ and a decoder $D$ to analyze and compress the CAD dataset. The dataset comprising sketch and extrude CAD models organized in a (S)olid-(P)rofile-(L)oop tree structure. This approach learns the distinct patterns inherent in the models by employing three discrete codebooks. Although the formats vary at each level, we have uniformly set the latent code length to 256 dimensions for ease of subsequent processing. Thanks to the powerful generative capabilities of our diffusion process, unlike HNC, we only need to use the simplest VQ-VAE without resid-

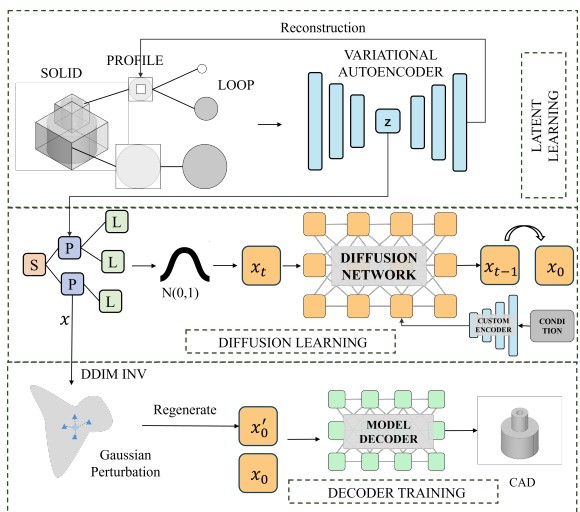

Figure 1: **Overview of CADiffusion.** We employ three distinct VQ-VAE models to perform data compression **(top)**. Building upon this, we represent CAD data as corresponding tree-structured latent representations, which serve as input to our diffusion model and can be guided by various inputs **(middle)**. To obtain a decoder suitable for CAD decoding, we designed specialized regularization strategies to ensure that samples from the Gaussian space generate reasonable CAD models **(bottom)**.

ual connections for modeling. Similar to hierarchical CAD representation, hierarchical codes are represented as a series of feature vectors, where each feature vector indicates a code or a separator token. We organize latent codes of three different levels into a tree structure, which is represented as follows: $[\text{S}, \langle\text{SEP}\rangle, \text{P}, \text{L}, \text{L}, \langle\text{SEP}\rangle, \text{P}, \text{L}, \text{L}, \text{L}, \text{L}, \langle\text{END}\rangle]$, where uppercase letters represent latent codes of the corresponding levels. The boundary command $\langle\text{SEP}\rangle$ indicates a new grouping of profile and loop codes, $\langle\text{END}\rangle$ indicates the end of the data. We pad zeros after the $\langle\text{END}\rangle$ to unify the length of different CAD models and then form two-dimensional tensors.

## 3.2 Diffusion Model for CAD

**Diffusion models (DMs)**: DMs learn a specific distribution by iteratively denoising a Gaussian variable through a fixed-length Markov chain, denoted as $T$. Specifically, given a data sample $x_0$ drawn from the distribution $q(x_0)$, two distinct processes are defined: a forward process $q(\boldsymbol{x}_{0:T})$, which progressively transforms a data sample into Gaussian noise, and a reverse process (generation process) $p_\theta(\boldsymbol{x}_{0:T})$, which gradually denoises the Gaussian noise back into the real data.

$$q(\boldsymbol{x}_{0:T}) = q(\boldsymbol{x}_0) \Pi_{t=1}^T q(\boldsymbol{x}_t \mid \boldsymbol{x}_{t-1}), \quad p_\theta(\boldsymbol{x}_{0:T}) = p(\boldsymbol{x}_T) \Pi_{t=1}^T p_\theta(\boldsymbol{x}_{t-1} \mid \boldsymbol{x}_t), \quad (1)$$

both $q(\boldsymbol{x}_t \mid \boldsymbol{x}_{t-1})$ and $p_\theta(\boldsymbol{x}_{t-1} \mid \boldsymbol{x}_t)$ represent Gaussian transition probabilities formulated as

$$q(\boldsymbol{x}_t \mid \boldsymbol{x}_{t-1}) = \mathcal{N}\left(\boldsymbol{x}_t; \sqrt{1-\beta_t}\boldsymbol{x}_{t-1}, \beta_t \boldsymbol{I}\right), \quad p_\theta(\boldsymbol{x}_{t-1} \mid \boldsymbol{x}_t) = \mathcal{N}\left(\boldsymbol{x}_{t-1}; \boldsymbol{\mu}_\theta(\boldsymbol{x}_t, t), \beta_t \boldsymbol{I}\right). \quad (2)$$

The mean variable $\boldsymbol{\mu}_\theta(\boldsymbol{x}_t, t)$ for the reverse transition $p_\theta(\boldsymbol{x}_{t-1} \mid \boldsymbol{x}_t)$ can be represented as:

$$\boldsymbol{\mu}_\theta(\boldsymbol{x}_t, t) = \frac{1}{\sqrt{\alpha_t}}\left(\boldsymbol{x}_t - \frac{\beta_t}{\sqrt{1-\bar{\alpha}_t}}\boldsymbol{\epsilon}_\theta(\boldsymbol{x}_t, t)\right), \quad (3)$$

where $\alpha_t = 1 - \beta_t, \bar{\alpha}_t = \Pi_{i=1}^t \alpha_i$, and $\beta_t$ gradually decreases to 0 as $t$ approaches 0. During the training stage of DMs, the evidence lower bound (ELBO) is maximized, eventually yielding the loss

function:

$$\mathcal{L}_{DM} = \mathbb{E}_{\boldsymbol{x},t,\boldsymbol{\epsilon}\sim\mathcal{N}(0,1)} \left[ \|\boldsymbol{\epsilon} - \boldsymbol{\epsilon}_\theta\left(\boldsymbol{x}_t,t\right)\|^2 \right] \tag{4}$$

In the process, $\boldsymbol{x}_t = \sqrt{\bar{\alpha}_t}\boldsymbol{x}_0 + \sqrt{1-\bar{\alpha}_t}\boldsymbol{\epsilon}$, $\boldsymbol{\epsilon}$ represents a noise variable and $t$ is uniformly sampled from the set $\{1,\ldots,T\}$. The key component of denoising diffusion models is the neural network-based score estimator $\boldsymbol{\epsilon}_\theta\left(\boldsymbol{x}_t,t\right)$, which serves as a time-step-conditioned denoising model.

**Diffusion Model for CAD**: Based on the description above, we organize tree logic latent representation into a 2-D tensor $z$ corresponding to CAD. Since we have already captured the organizational logic of CAD in the tree-latent space, we aim to use the diffusion model to fit the data on a holistic level and generate coherent CAD models. Specifically, we obtain $z_t, t \in \{1,\ldots,T\}$ from a sample $z_t$ by incrementally introducing Gaussian noise with a predetermined variance schedule. Subsequently, we employ a transformer-based time-conditional denoising model $\epsilon_\theta$. To train the denoising model, we utilize the simplified objective introduced by Ho *et al.* Ho et al. (2020) :

$$L_{\text{simple}}\left(\theta\right) := \mathbb{E}_{z,\epsilon\sim N(0,1),t} \left[ \|\epsilon - \epsilon_\theta\left(z_t,t\right)\|^2 \right]. \tag{5}$$

During the inference phase, we generate $\widehat{z}_0$ by progressively removing noise from a variable sampled from the standard normal distribution $N(0,1)$.

## 3.3 CAD DECODER REGULARIZATION TERM

The unique challenges presented by our latent diffusion model, which generates hierarchical, tree logic latent representations, are presented below. Parsing generated latents into different hierarchies is prone to errors and can be overly cumbersome. Instead, the latents generated encapsulate the logic and components of CAD, enabling direct decoding into a CAD model. This approach is indeed more efficient; however, it is crucial to ensure its accuracy as well. Although the three different levels of latents is logically assembled together, decoding it into a realistic CAD model is more challenging than reconstructing a single level of CAD components separately. We found that training this decoder solely with the latents of training dataset is insufficient and leads to some unrealistic decoded results. To enhance the stability and fidelity of our CAD generation process, a novel regularization technique is developed. This technique involves perturbing the latent space to simulate variations that the diffusion model might generate, thus training the decoder to be resilient to these variations and ensuring smoother transitions between different CAD models. The regularization process consists of several steps, detailed below:

**Inverse Mapping to Noise Space:** Initially, the latent representation from dataset is mapped back to the noise space using the DDIM inversion method Song et al. (2021). The DDIM inversion process systematically reintroduces noise into a clean latent representation to reach a noised state that can then be diffused to regenerate the original latent, effectively serving as a way to explore variations in the generated CAD models. Starting from a latent at the initial time step, the DDIM inversion aims to compute a corresponding noised latent after T steps. The inversion process is governed by the following equation:

$$\hat{z}_t = \sqrt{\alpha_t}\,\frac{\hat{z}_{t-1} - \sqrt{1-\alpha_{t-1}}\varepsilon_\theta}{\sqrt{\alpha_{t-1}}} + \sqrt{1-\alpha_t}\varepsilon_\theta, \tag{6}$$

where $\alpha_t$ is a pre-determined noise schedule parameter, and $\varepsilon_\theta$ is the neural network predicting the noise component.

**Gaussian Perturbation:** Once the DDIM inversion maps the clean latent $z_0$ to a noised latent $\hat{z}_T$, we apply Gaussian perturbation as part of our regularization strategy. The perturbed noise latent $\hat{z}'_T$ is then given by:

$$\hat{z}'_T = (1-\sigma)\hat{z}_T + \sigma\mathcal{N}(0,I), \tag{7}$$

where $\sigma$ is a scaling factor and $\mathcal{N}(0,I)$ represents isotropic Gaussian noise. The perturbed noise vector is then used to regenerate a new latent vector $\hat{z}'_0 = DDIM(\hat{z}'_T)$ through the forward diffusion process.

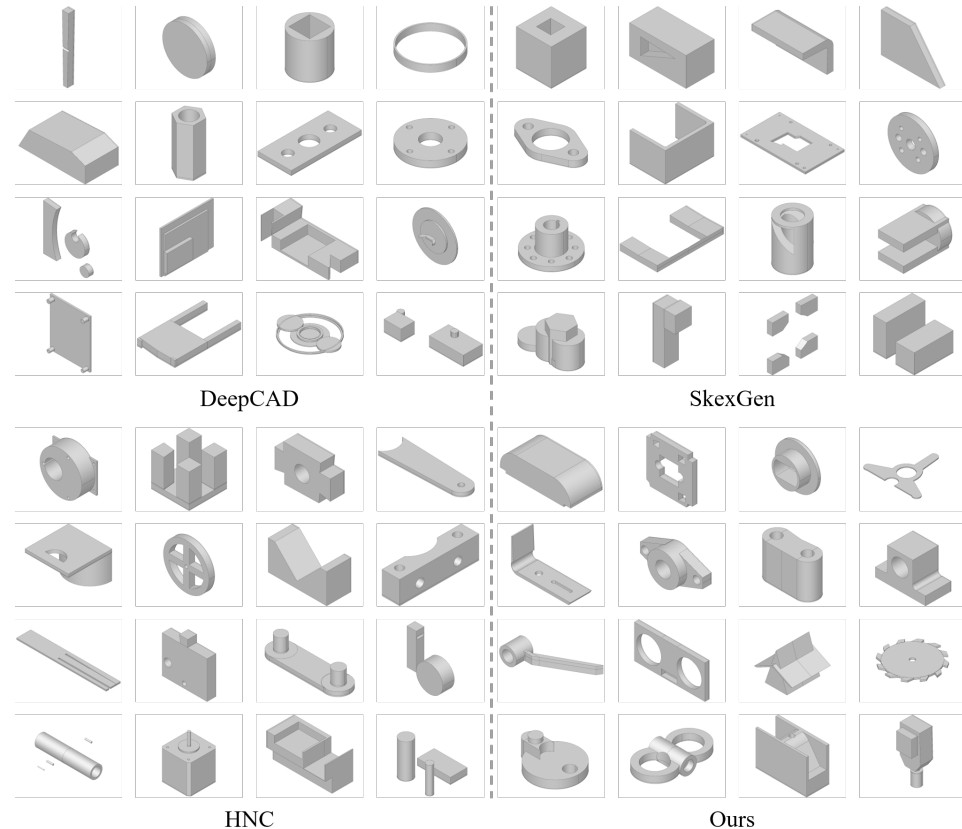

Figure 2: **Unconditional generation results of four different methods.** DeepCAD and SkexGen frequently resort to assembling simple components, resulting in CAD models that lack the rationality. While HNC's outcomes display significant improvement, they occasionally contain artifacts with small components. In contrast, our method produces high quality results with well structure.

**Distance Minimization:** The perturbation does not deviate far enough from the original latent, we can minimize the distance between the decoded results of the perturbed latents and the original CAD:

$$\min_D \|D(\hat{z}'_0) - CAD\|. \tag{8}$$

This regularization term aims to train the decoder to produce smooth and consistent CAD models, reducing artifacts, and ensuring that the models are robust to variations in the latent input. The latent representations $z_0$ of the original data are also used to train this decoder. This enhanced approach not only addresses the complexity of translating hierarchical latent structures into functional CAD designs but also significantly improves the adaptability and quality of the generated models.

## 3.4 CONDITIONAL GENERATION

The ability to randomly sample shapes offers limited scope for interaction, underscoring the importance of learning a conditional distribution for user applications. It is crucial to accommodate multiple forms of conditional inputs to address diverse scenarios effectively. Using the flexible conditional mechanism facilitated by the diffusion model, we integrate multiple conditional input modalities using task-specific encoders $E_\phi$ and a cross-attention module. To enhance flexibility in controlling the distribution, we adopt classifier-free guidance for conditional generation. For training such conditional model, the objective function is formulated as follows:

$$L\left(\theta, \{\phi_i\}\right) := \mathop{\mathbb{E}}_{z, \mathbf{c}, \epsilon, t}\left[\left\|\epsilon - \epsilon_\theta\left(z_t, t, D \circ E_{\phi_i}\left(\mathbf{c}_i\right)\right)\right\|^2\right] \tag{9}$$

The task-specific encoder $E_{\phi_i}(\mathbf{c}_i)$ is employed for the $i^{\text{th}}$ modality, while $D$ represents a dropout operation facilitating classifier-free guidance. In this work, we mainly explore two conditions with many practical applications: using point clouds and initial user input.

## 4 EXPERIMENTS

### 4.1 IMPLEMENTATION DETAILS

**Dataset**: Using the extensive DeepCAD dataset Wu et al. (2021), we acquire ground truth sketch-and-extrude models, comprising 178,238 instances. These models are divided into a training set (90%), a validation set (5%), and a test set (5%). To ensure the integrity of the data set, we implement methods similar to previous studies Willis et al. (2021c); Xu et al. (2022) to detect and eliminate duplicate models from the training set. In addition to removing duplicate models, we extract hierarchical properties for loops, profiles, and solids, and subsequently remove duplicate properties at each level. Furthermore, for training purposes, CAD models are included only if they meet specific criteria: a maximum of 5 solids, 20 loops per profile, 60 curves per loop, and a maximum of 200 commands in the sketch-and-extrude sequence. Following the duplicate removal and filtering processes, the training dataset comprises 102,114 solids, 60,584 profiles, and 150,158 loops for codebook learning. Additionally, 124,451 sketch-and-extrude sequences are retained for CAD model generation training. For CAD engineering drawings, we adopt the approach described in SkexGen Xu et al. (2022) and extract sketches from DeepCAD. A total of 99,650 sketches are utilized for training purposes after duplicate removal.

**Other Details:** The model is trained on a Nvidia RTX A100 GPU with a batch size of 256. Each VQ-VAE model and model decoder are trained for 250 epochs. For the randomly generated and conditionally generated diffusion models, we train them for 350 epochs and 500 epochs, respectively. We use the AdamW optimizer Loshchilov & Hutter (2018) with a learning rate of 0.001 after a linear warm-up for the first 2000 steps. The VQ-VAE network consists of 4 layers. For the diffusion model, there are six blocks, each comprising a self-attention layer and a fully-connected layer. If it is a conditional generation, each block also includes a cross-attention layer. During the generation process, we use DDIM for sampling, with a sampling step of 100 steps. For the corresponding scale of the CFG, we set it to 3. $\sigma$ for perturbation is set it to 0.1. More details can be found in the appendix.

**Evaluation Metrics:** We use five established metrics to quantitatively assess random generation. Three metrics are based on point clouds sampled on the model surfaces. Two metrics scrutinizing generated tokens originating from sketch and extrude construction sequences. For point-cloud evaluation, 2,000 points are sampled from each generated and ground-truth dataset, facilitating a comparative analysis of the two sets. Descriptions of metrics are referred to in the appendix.

### 4.2 UNCONDITIONAL GENERATION

For the unconditional generation, we conduct comparisons with three CAD generation works, DeepCAD, SkexGen and HNC. The results of the other three methods were obtained using publicly available code, and we used their default settings. Each method produces 10,000 CAD models, which are then compared with a randomly selected subset of 2,500 ground truth models from the test set. We compute all metrics three times and take the average for comparison.

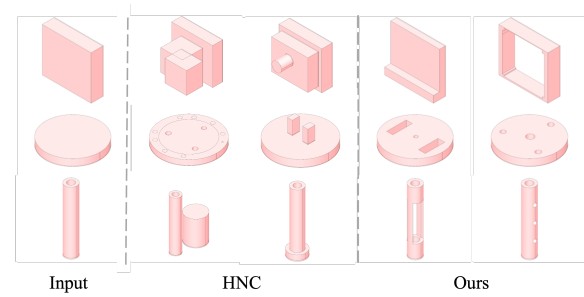

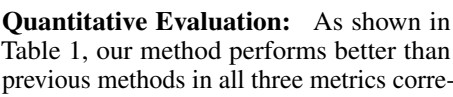

Figure 3: **Conditional generation results of CAD from initial user input.**

**Quantitative Evaluation:** As shown in Table 1, our method performs better than previous methods in all three metrics corresponding to point clouds, especially MMD and JSD, which are significantly better than the baseline methods, demonstrating notable improvements in both quality and diversity. The uniqueness score of our method is similar to the previous two methods and significantly better than DeepCAD. Although SkexGen's novelty score is similar to ours, it fails to generate highly complex CAD models (see

Table 1: Quantitative evaluations on the CAD generation task based on the Coverage (COV) percentage, Minimum Matching Distance (MMD), Jensen-Shannon Divergence (JSD), the percentage of Unique, Novel scores and Realism.

| Method | COV % ↑ | MMD ↓ | JSD ↓ | Novel % ↑ | Unique % ↑ | Realism % ↑ |
|---|---|---|---|---|---|---|
| DeepCAD | 79.98 | 1.21 | 3.34 | 90.5 | 86.4 | 36.4 |
| SkexGen | 83.58 | 1.11 | 0.91 | 99.2 | 99.8 | 42.3 |
| HNC | 86.62 | 1.03 | 0.74 | 94.1 | 99.7 | 44.1 |
| Ours w/o reg | 89.03 | 0.12 | 0.13 | 99.8 | 99.7 | 43.1 |
| Ours | 90.08 | 0.10 | 0.13 | 99.8 | 99.6 | 51.3 |

Table 2: Comparison with DeepCAD and Draw Step by Step, mean and median Chamfer Distance (CD) results. By employing more advanced conditional generation models, our method obtains more accurate reconstruction results.

| Model | Mean CD ↓ $(\times 10^3)$ | Median CD ↓ $(\times 10^3)$ |
|---|---|---|
| DeepCAD | 43.18 | 9.836 |
| Draw Step by Step | 39.16 | 7.821 |
| Ours | **32.17** | **6.304** |

Figure 2), as reported in previous methods. The comparison of these results demonstrates that our distribution fitting is quite effective. However, these metrics are intended to indicate how closely the model's output matches the real distribution. They do not adequately measure the realism of the results. To better demonstrate the effectiveness of our approach from a quantitative perspective, we introduce a Human Evaluation similar to that in HNC Xu et al. (2023) to measure the realism of the generated complex results. For specific practices, please refer to the appendix. From this realistic comparison result in Table 1, it can further be seen that our method is capable of learning to generate complex and realistic models.

**Qualitative Evaluation:** As shown in Figure 2, the results of DeepCAD do not exhibit significant issues when generating a simple CAD. However, when tasked with generating complex structures, it often resorts to assembling original components and struggles to create CADs that resemble real-world examples in a rational manner. SkexGen may perform slightly better than DeepCAD, but it still encounters similar challenges. HNC's results show considerable improvement, yet artifacts with small components are occasionally present. In comparison, our approach yields results with more well-structured characteristics, closely resembling real mechanical parts.

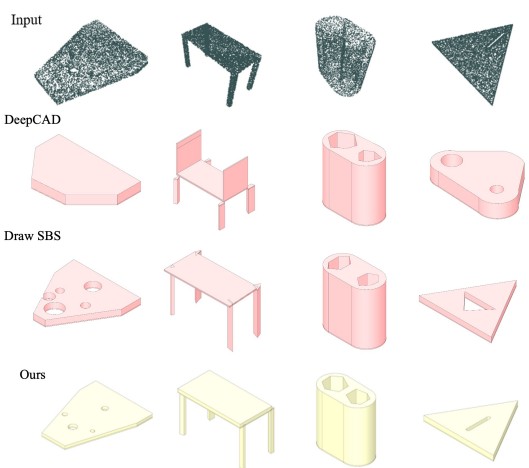

Figure 4: **Conditional generation results of CAD from point clouds.**

### 4.3 CONDITIONAL GENERATION

**Autocompletion from User Input**: We consider generating a detailed model given an initial model, and we can also use this type of input as control conditions for the diffusion model to generate corresponding potential CAD models for automatic completion purposes. During the training process, we utilize random initial inputs as conditions for the conditional encoder, which serves as input for the diffusion model to reconstruct the corresponding complete CAD models. The encoder encodes the extruded profile parameters and shares the same structure as the encoder employed in training the codebook with VQ-VAE. Figure 3 shows the corresponding CAD autocomplete results with rich details from initial extruded profiles. Each row contains multiple generated results, each corresponding to different noise samples. It can be observed that the autocomplete results are generally reasonable and of high quality, which can assist

designers in CAD design. HNC has also implemented a similar functionality, and we have compared it. Due to the lack of a more suitable comparative method, we also employed human evaluation to test Realism. The results for the HNC Realism were 48.2%, while ours were 52.5%.

**Point to CAD**: 3D reverse engineering entails inferring a CAD model from a 3D scan, a process that demands the expertise of designers and often consumes considerable time. Our method can control the generation process through conditional input from point clouds to obtain CAD models that closely resemble the point clouds, thus achieving a relatively rapid and accurate reverse engineering process to some extent. The encoder used here is a pre-trained ULIP Xue et al. (2023) PointNet Qi et al. (2017). Next, we assess the proposed method for CAD generation based on the point cloud condition. As shown in Figure 4, given a point cloud, our method can obtain a CAD model that is generally similar in overall structure. DeepCAD has also implemented a similar functionality, and we have compared it with our method and another method called Draw Step by Step Ma et al. (2024). For the reconstructed CAD results, we assess them quantitatively against ground truth CAD models using mean and median Chamfer Distances (CD). The quantitative comparison results are shown in Table 2.

### 4.4 ABLATION STUDY

To demonstrate the significance of the regularization term we introduced, we conducted a comparative analysis between models with and without this regularization. As illustrated in Figure 5, the decoder enhanced with regularization is capable of rationalizing outputs that were previously deemed unrealistic. Initially, some of these unsatisfactory results were attributed to noise and a lack of inherent symmetrical logic in the decoded outputs. By integrating a smoothness-inducing regularization, we observed that the decoder could produce CAD models that more closely align with the intrinsic logic of mechanical parts.

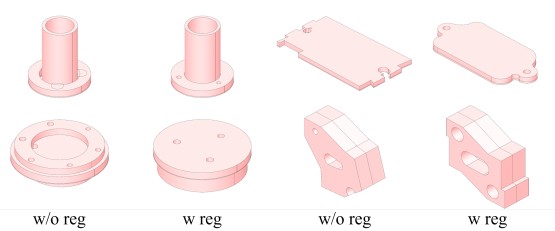

w/o reg    w reg    w/o reg    w reg

Figure 5: **Ablation study.** It can be observed that the results on the right, which incorporate regularization, are more aesthetically pleasing and logical, with less noise compared to those without regularization. For more visual results, please refer to the appendix.

As shown in the last two rows of Table 1, after adding the regularization terms, the realism of the results generated by our method has significantly improved. Our regularization improves other metrics as well, but not as significantly, because, as with HNC, the Realism metric measures the complex results with three or more extrusions, as only such complex results have evaluative value. Other metrics measure the average across all generated results, and our regularization does not significantly improve simple CAD results, which have limited potential for improvement. This leads to a relatively minor improvement in other metrics when averaged. No single metric can fully evaluate the quality of generated results, generating complex but unrealistic results can lead to high Novelty and Uniqueness scores, without significantly affecting other metrics that measure the diversity of generated results.

## 5 CONCLUSIONS AND FUTURE WORK

In conclusion, we have introduced CADiffusion, a novel diffusion-based generative model tailored for Computer-Aided Design (CAD) data generation. Our approach addresses the persistent challenges in producing diverse and high-quality CAD shapes by seamlessly integrating diffusion models and Vector Quantized Variational Autoencoders (VQVAE) to obtain latent representations. Through extensive experimentation, we have demonstrated the effectiveness of CADiffusion, achieving state-of-the-art performance on benchmark datasets. Additionally, we introduced a regularization method specifically tailored for training the decoder. This method employs perturbations in the Gaussian space to smooth the decoder's outputs, thereby enabling it to produce more reasonable results. We believe that CADiffusion opens up new possibilities for advancing 3D shape generation in practical CAD modeling and design applications.

**limitation.** The current results of point to CAD do not fully match the input yet. In the future, we hope to explore better reverse engineering methods using diffusion priors.

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

## A    EVALUATION METRICS

Due to the unique characteristics of CAD data modalities, we have provided detailed explanations of the five different metrics mentioned in the main paper here, to facilitate reader understanding.

- Coverage (COV) denotes the proportion of ground-truth models that contain at least one matched generated sample, where matching is determined based on the Chamfer distance (CD) or Earth Mover's distance (EMD). COV serves as a measure of the diversity of generated shapes, revealing potential mode collapse if only a few ground-truth models are matched, resulting in low coverage scores.

- Minimum Matching Distance (MMD) calculates the average minimum matching distance between the ground truth and generated sets.

- Jensen-Shannon divergence (JSD) assesses the similarity between two probability distributions, reflecting the degree to which ground truth and generated point clouds occupy similar locations. Utilizing voxelization, occupancy distributions are computed to derive the JSD score.

- Novelty indicates the percentage of generated CAD sequences absent in the training set, while uniqueness signifies the percentage of generated data that appear once within the generated set.

## B    INPLEMETATION DETAILS

The tree structure corresponding to the CAD is obtained directly by parsing according to the CAD logic, implemented by a segment of code. This part is the same as in HNC. The CAD latent used for diffusion learning is in the form of tensors with a shape of $32 \times 256$, the input and output shapes are the same. Therefore, all our model architectures use 1D transformer structures, which mainly include Self-Attention, Fully Connected Layers, and Cross-Attention layers. In the Self-Attention mechanism, the basic QKV form corresponds to different matrix transformations followed by attention calculations. In the Cross-Attention layer, the input conditions are also transformed into an $n \times 256$ format by a specific encoder and then used as the KV components in the attention computation.

## C    HUMAN EVALUATION

Our approach to Human Evaluation is similar to HNC and SolidGen. For each method that need to be evaluated, we randomly select models with three or more extrusions from their results generated without any conditions. For each model created by a generation method, we randomly choose a real model from the dataset and display the rendered images of these two models side by side. We randomly select 100 results from each method, and the image pairs are presented to college participants, who are asked to assess which one appears more "realistic."

## D    MORE EXPERIMENTAL RESULTS

Here, we include visualizations of more results. Randomly generated results are shown in Figure 6, autocompletion results are shown in Figure 7, and Point2CAD results are shown in Figure 8. For more ablation study results, please refer to Figure 9.

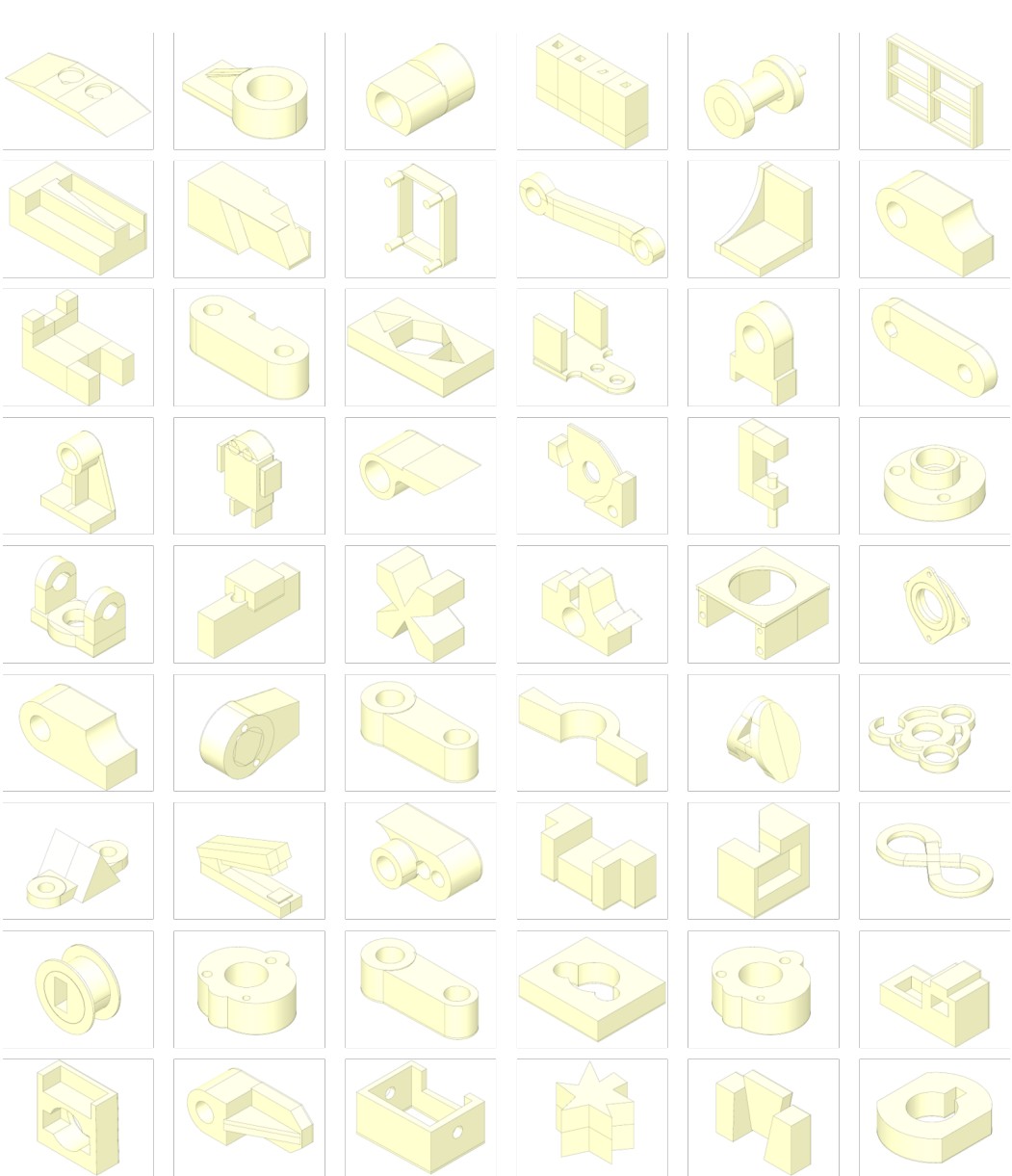

Figure 6: **Random generation results of CADiffusion.** The CADiffusion model can generate diverse and realistic CAD models.

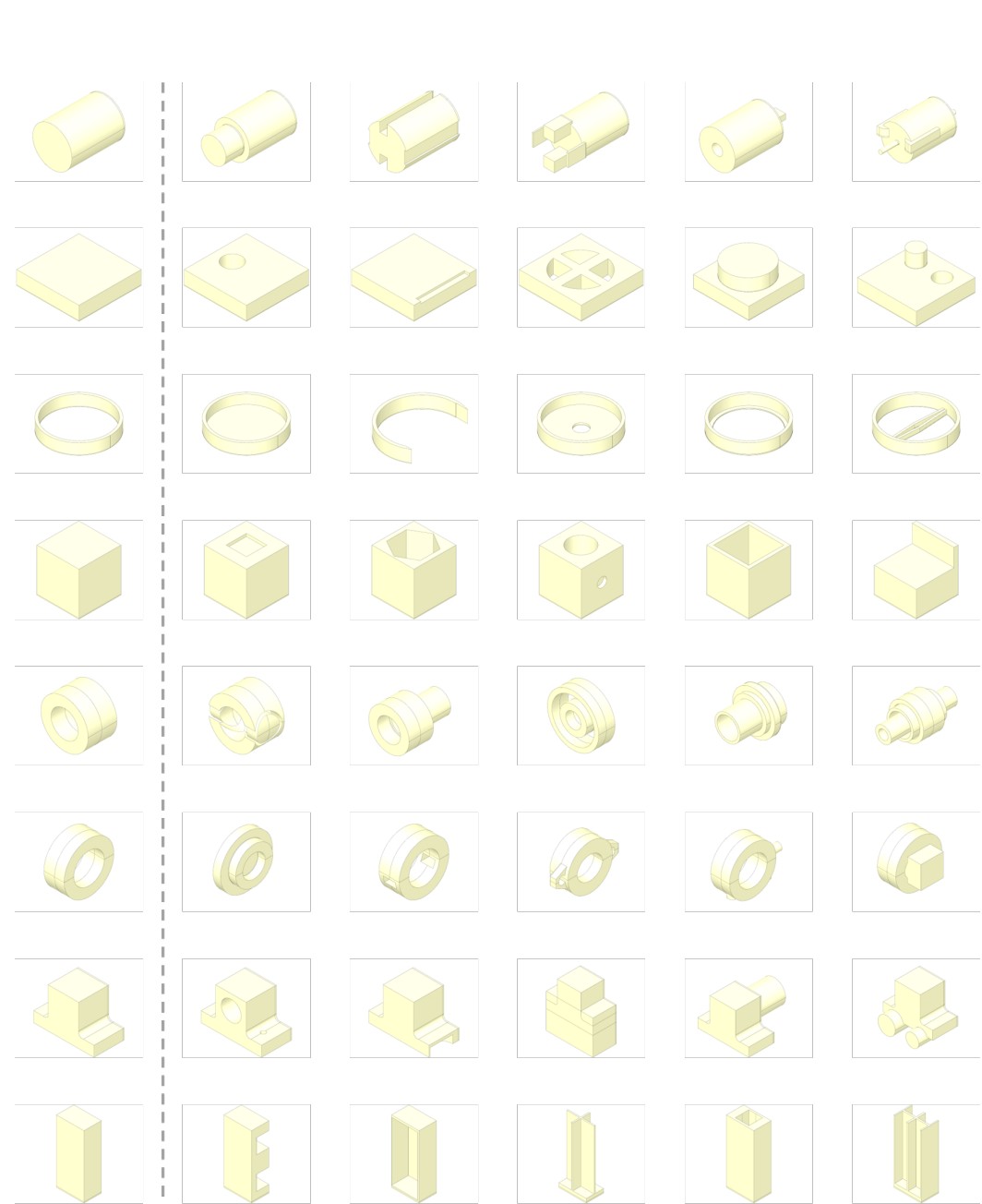

Figure 7: **Conditional generation results of CAD from initial user input.** The leftmost column represents initial conditional inputs. The autocomplete results demonstrate a consistent level of quality and reasonableness, providing valuable assistance to CAD designers in their design processes.

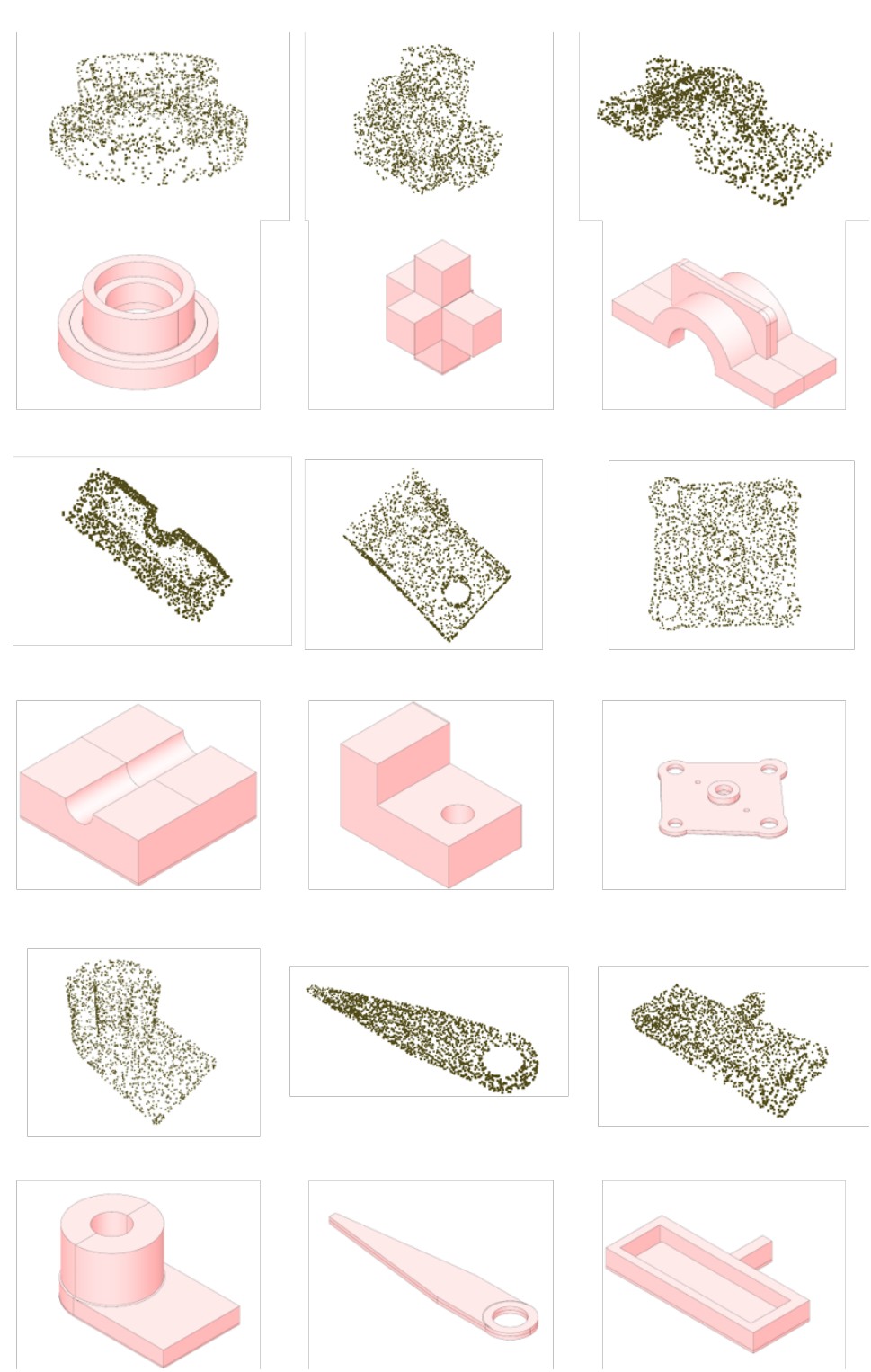

Figure 8: **Conditional generation results of CAD from point clouds.** The rows from top to bottom consist of input point clouds and their corresponding generated CAD models. Based on a provided point cloud, our approach is capable of deriving a CAD model that exhibits a broad similarity in its overall structure.

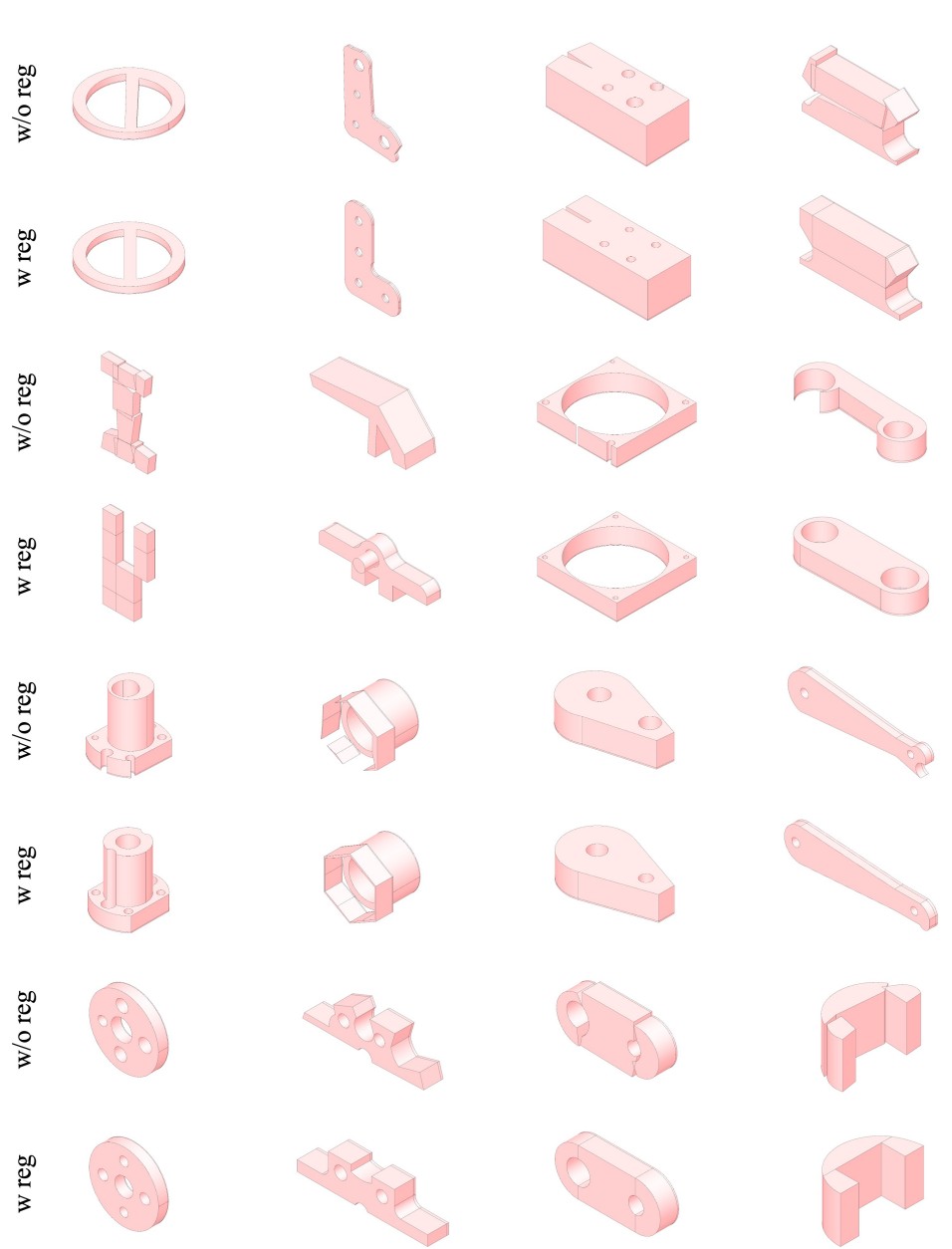

Figure 9: **Ablation studies.** It can be observed that the results which incorporate regularization, are more aesthetically pleasing and logical, with less noise compared to those without regularization.

