# OpenReview forum: "Regularized Diffusion Modeling for CAD Representation Generation"
_ICLR.cc/2025/Conference — ICLR 2025 Conference Withdrawn Submission_

### Official Review · Reviewer_TYJy · 2024-10-17

**Soundness:** 2
**Presentation:** 3
**Contribution:** 2
**Rating:** 5
**Confidence:** 4

**Summary:**

This paper introduces a diffusion-based method with the designed regularization technique to for generating CAD sequences. The main cores are centered in two points: 1) conducting the latent diffusion to denoise the hierarchical code tree (a kind of representation of CAD command sequence). 2) specifying a decoder with the regularization term to enhance the stability and fidelity of the diffusion process. The authors simultaneously report unconditional and conditional tasks of (e.g., from point cloud) CAD generations under several metrics. The promising results demonstrated the effectiveness of their proposed framework.

**Strengths:**

S1: This paper focus on an interesting problem that can effectively serve the practical needs of modern engineering and manufacturing.

S2: This paper is well organized and easy to follow.

S3: The promising results under several metrics are significantly surpasses previous efforts.

S4: Conditional and unconditional CAD sequence generations are very practical that have great potential to promote the industrial development.

S5: Integrating the diffusion model into the CAD sequences generation is reasonable and effective (based on the experimental results).

**Weaknesses:**

W1: The claim “We are the first to explore the use of diffusion models for CAD generation, and have designed corresponding models and representations” is clearly not rigorous. First, the representation of CAD used (hierarchical code tree) in the manuscript is very similar to the previous effort (HNC-CAD). Second, VQ-CAD is also a pipeline that conducts diffusion method in CAD generation (just name a few). Unfortunately, this would weaken the innovation of proposed framework. The reviewer would suggest the authors clearly point out the differences from them in the related works section.

W2: According to the Table 1, it seems that the improvement is mainly brought by diffusion model. When comparing w/o reg and w/ reg, most metrics are identical or just slight difference, which indicates the regularization in the decoder that only gives marginal positive effort. Besides, the designed regularization only brings the positive feedback under Realism metric but other metrics are almost unchanged, which does not make any sense. Overall, this paper lacks of detail analysis of this reported result, which is quite important as the regularization decoder plays a crucial role in the proposed framework claimed by authors.

W3: The metric used in unconditional generation is insufficient, making it hard to prove the capability of the proposed method solidly. For the CAD sequence generation, another important perspective is to measure its validity. The reviewer highly recommends the authors report the invalid ratio of the generated CAD sequences.

W4: At present, the CAD models showcased in the manuscript are relatively simple, which still has a significant gap compared to the industrial applications. It would be better to provide more visualizations of complex CAD models (e.g., more types of CAD commands and longer CAD sequences).

The most concern is that, the technical sound seems to be performing poorly as the proposed regularization part contributes marginally.

**Questions:**

Q1: Would please authors clarify the differences between HNC-CAD and proposed method (CAD representation)?

Q2: How many samplings have author's done (per CAD model) when measuring three metrics related to point clouds?

Other questions are listed in the Weaknesses, please refer to W1-W4.
The reviewer may adjust the score based on the authors’ feedback.

---

### Official Review · Reviewer_NPbw · 2024-11-02

**Soundness:** 2
**Presentation:** 1
**Contribution:** 2
**Rating:** 3
**Confidence:** 4

**Summary:**

This paper proposes CADiffusion, a diffusion-based generative model for Computer-Aided Design (CAD) model generation. It is focused on producing high-quality and diverse CAD shapes and flexible conditional control. The model uses a hierarchical latent representation and a specialized decoder with regularization strategies. The contributions include exploring diffusion models for CAD generation and introducing a new regularization strategy.

**Strengths:**

+ CAD has wide applications in various industries, and the ability to generate high-quality and diverse CAD models is crucial for innovation in production processes.
+ The use of diffusion models for CAD generation is somewhat novel.

**Weaknesses:**

- The paper may benefit from reorganizing the structure by moving the introduction of diffusion models and DDIM inversion to related work and preliminaries, so the main proposed method is more prominent.
- The main idea of the paper is to apply Diffusion model into CAD generation, which is not contributive enough. The main innovation of the paper may be the regularization of eq 8 and the controllability in eq 9. These are not innovative, too.
- As in table 1, the benefit gained with the regularization seem to be trivial in most metrics.
- CAD is mostly used in industrial design. The main evluation should be from precise control. However, the reviewer is confused with Fig 5 where the proposed method is discussed from the perspective of aesthetic and logic.
- The definition of "realism" in the context of CAD models could be more precise. The current evaluation of realism seems to be based on a comparison with ground truth models, but a more detailed breakdown of what constitutes realism in CAD, such as geometric accuracy, topological correctness, and compliance with engineering standards, would provide a more comprehensive understanding.

**Questions:**

Please see weakness.

---

### Official Review · Reviewer_t6AM · 2024-11-04

**Soundness:** 3
**Presentation:** 2
**Contribution:** 2
**Rating:** 5
**Confidence:** 3

**Summary:**

The paper contributes a diffusion-based generative model, CADiffusion, to generate diverse CAD models. The method utilizes a hierarchical CAD representation (to model the inherent CAD logic of sequential operations) and proposes combining diffusion models and VQVAEs to generate latent representations. To capture the CAD logic effectively, they introduce a novel regularization strategy that perturbs the latent space to make it resilient to variations. Their quantitative evaluations indicate better performance than previous methods.

Overall, this paper could be a significant contribution to the generative CAD area, provided certain clarifications on broader evaluations are addressed.

**Strengths:**

1. Motivation for diffusion-based CAD generation is strong.

2. The regularization strategy is a smart move and is one of the major reasons (if not the most) for increased performance.

**Weaknesses:**

1. Conditional generation of CAD from point cloud is a limitation (as noted by the authors).

2. The authors mention using an NVIDIA A100 GPU, which suggests that substantial computational resources are required and does not give enough confidence in the feasibility of this method's adoption into CAD workflows.

3. Filtering the dataset based on criteria (as mentioned by the authors in sec 4.1) might limit CADiffusion's generalizability.

**Questions:**

1. How might one go about integrating this framework into an actual CAD workflow?

2. For conditional point cloud-based generation, have the authors considered comparing against Point2CAD (Liu, Yujia, et al. "Point2CAD: Reverse Engineering CAD Models from 3D Point Clouds." Proceedings of the IEEE/CVF Conference on Computer Vision and Pattern Recognition. 2024.)?

3. Given that the dataset only considers "...maximum of 5 solids, 20 loops per profile, 60 curves per loop and a maximum of 200 commands..." how can the current framework be extended to generate complex CAD assemblies?

4. What are the inference times for conditional/unconditional generation?

5. Have other filtering techniques been explored for training purposes? If so, how do they impact performance in terms of the metrics? (This is in reference to sec 4.1.)

---

### Official Review · Reviewer_jyfF · 2024-11-04

**Soundness:** 2
**Presentation:** 3
**Contribution:** 2
**Rating:** 5
**Confidence:** 3

**Summary:**

This paper presents a diffusion-based model for generating 3D CAD models. The proposed approach employs three VQ-VAE modules to create a hierarchical latent representation that aligns closely with the CAD design workflow. Additionally, the authors introduce a regularization strategy that introduces variance in the latent space, enhancing the stability of the decoder's output.

**Strengths:**

This paper presents an interesting approach to 3D model generation using hierarchical latent representation. The proposed framework, along with its regularization strategy, enables the generation of more realistic CAD models.

**Weaknesses:**

1. Although the author claims to be the first application of diffusion models in 3D CAD model generation, the problem of hierarchical 3D generation by diffusion models have been studied in many research works such as [1] [2] [3], where the problem is more complicated.
2. For researcher who are not working on CAD model design, the work could be easier to understand if additional background information on CAD model design, specifically regarding the challenges inherent in this field, could be provided. Providing context on issues such as complexity in shape representation, scalability of model designs, and accuracy in high-dimensional spaces would strengthen the reader's understanding of the problem the proposed method aims to address.
3. In Figure 2, the generated 3D shapes appear relatively simple, raising concerns about the method's applicability to more complex 3D model generation tasks. Including evidence or examples that demonstrate the model’s capability to handle intricate designs or more varied geometries would provide a clearer indication of the approach's scalability and versatility.
4. The paper relies primarily on subjective evaluations and lacks objective assessments of the generated structures. For instance, a quantitative comparison between human-designed CAD models and models generated by the proposed framework could offer a more rigorous evaluation. Incorporating metrics such as structural accuracy, fidelity to design intent, or user satisfaction would enable a more comprehensive assessment of the model’s output quality.
5. The author misses energy-based models in 3D points/shape generation, such as [4][5], in discussion and comparison of related works and experiments.

[1] Sun, Jingxiang, et al. "Dreamcraft3d: Hierarchical 3d generation with bootstrapped diffusion prior." arXiv preprint arXiv:2310.16818 (2023).

[2] Kim, Seung Wook, et al. "Neuralfield-ldm: Scene generation with hierarchical latent diffusion models." Proceedings of the IEEE/CVF conference on computer vision and pattern recognition. 2023.


[3] Po, Ryan, and Gordon Wetzstein. "Compositional 3d scene generation using locally conditioned diffusion." 2024 International Conference on 3D Vision (3DV). IEEE, 2024.

[4] Xie, Jianwen, et al. "Generative pointnet: Deep energy-based learning on unordered point sets for 3d generation, reconstruction and classification." Proceedings of the IEEE/CVF Conference on Computer Vision and Pattern Recognition. 2021.

[5] Xie, Jianwen, et al. "Learning descriptor networks for 3d shape synthesis and analysis." Proceedings of the IEEE conference on computer vision and pattern recognition. 2018.

**Questions:**

1. Could the author explain what is the application of CAD model generation, especially for unconditional generation?
2. What is the difference between 3D CAD model generation with other conditional generation methods, like compositional 3D scene generation?
3. Could the author give more details about how human evaluation is conducted?
4. Is there any comparison with text-based generation methods?
5. Except for human subjective evaluation, is there any quantification method for rationality evaluation of the generations results?It is hard to tell the difference from Figure 2 for people who don't work on CAD modeling.

---

### Official Review · Reviewer_kQ5r · 2024-11-04

**Soundness:** 3
**Presentation:** 3
**Contribution:** 1
**Rating:** 3
**Confidence:** 5

**Summary:**

Authors proposed a new diffusion model for generating the CAD sketch-and-extrude. A diffusion-based regularization strategy is also used to ensure improve the fidelity of the decoded result. Results improved for multiple metrics compared to baseline.

**Strengths:**

Paper is well-written and easy to follow. The proposed regularization method based on DDIM inversion is simple, intuitive, and improve the overall results. This help to address the additional noise issue been introduced by switching to diffusion.

**Weaknesses:**

The proposed model is heavily based on HNC-CAD with similar data representation and encoding strategy. Main difference is the use of diffusion to generate flattened CAD tokens as opposed to autoregressive. The proposed regularization term also looks like the approach used in “Cascaded diffusion models for high fidelity image generation”, where some amount of noise is added to ground-truth. Difference is that a DDIM inversion is used here instead. So while the performance improvement is adequate, method contribution is limited. Also there is no closest-retrieved analysis for checking that model is not simplify remembering training data. And I believe this paper is not the first to use diffusion model to generate CAD. There are other earlier works e.g. VQ-CAD Computer-Aided Design model generation with vector quantized diffusion. While the model been used is different (vq-cad used discrete diffusion), some of the claims could still be revised to be more accurate.

**Questions:**

If possible, it will be great to include an ablation study for the closest-retrieved cad from the training set. This could be either based on chamfer distance or light field distance (as in fig.6 of Neural Wavelet-domain Diffusion for 3D Shape Generation). Without this, it is very hard to determine the ability of the model to generate novel shapes as opposed to memorizing training data.

I am also curious abou the failure rate. Seems like point-based metrics are computed over all generated shapes. But how many of those shapes are actually be build into valid CAD breps?

---

### Note · Authors · 2024-11-12

I have read and agree with the venue's withdrawal policy on behalf of myself and my co-authors.